# Acceleration of Electrospun PLA Degradation by Addition of Gelatin

**DOI:** 10.3390/ijms24043535

**Published:** 2023-02-10

**Authors:** Alexandra Bogdanova, Elizaveta Pavlova, Anna Polyanskaya, Marina Volkova, Elena Biryukova, Gleb Filkov, Alexander Trofimenko, Mikhail Durymanov, Dmitry Klinov, Dmitry Bagrov

**Affiliations:** 1School of Biological and Medical Physics, Moscow Institute of Physics and Technology (National Research University), 141701 Dolgoprudny, Russia; 2Lopukhin Federal Research and Clinical Center of Physical-Chemical Medicine of Federal Medical Biological Agency, 119435 Moscow, Russia; 3Department of Radiochemistry, Faculty of Chemistry, Lomonosov Moscow State University, 119991 Moscow, Russia; 4Department of Bioengineering, Faculty of Biology, Lomonosov Moscow State University, 119991 Moscow, Russia

**Keywords:** polylactide, gelatin, electrospinning, nanofibers, mechanical properties, degradation, biocompatibility

## Abstract

Biocompatible polyesters are widely used in biomedical applications, including sutures, orthopedic devices, drug delivery systems, and tissue engineering scaffolds. Blending polyesters with proteins is a common method of tuning biomaterial properties. Usually, it improves hydrophilicity, enhances cell adhesion, and accelerates biodegradation. However, inclusion of proteins to a polyester-based material typically reduces its mechanical properties. Here, we describe the physicochemical properties of an electrospun polylactic acid (PLA)–gelatin blend with a 9:1 PLA:gelatin ratio. We found that a small content (10 wt%) of gelatin does not affect the extensibility and strength of wet electrospun PLA mats but significantly accelerates their in vitro and in vivo decomposition. After a month, the thickness of PLA–gelatin mats subcutaneously implanted in C57black mice decreased by 30%, while the thickness of the pure PLA mats remained almost unchanged. Thus, we suggest the inclusion of a small amount of gelatin as a simple tool to tune the biodegradation behavior of PLA mats.

## 1. Introduction

Biodegradable polymers are used in bone [1], cartilage [2], nerve [3,4], vessel [5,6], and tendon [7] tissue engineering. Polyesters have proven to be both durable and extensible biodegradable polymers for production of matrices and implants. Polylactide (PLA) is a polyester of lactic acid, a weak organic acid, that is nontoxic to biological systems [8,9,10]. PLA, as well as PLA-based composites and copolymers, are widely used in surgery (sutures, meshes, orthopaedical devices), packaging, textiles, and other applications. The mechanical properties of PLA strongly depend on its stereochemistry—the ratio of L- and D-lactic acid monomers, and their positions in the polymer chain [11,12].

PLA is structurally the simplest member of the biomedical polyester family, which includes polydioxanone, poly (glycolic acid), poly (3-hydroxybutyrate), and other polymers [12]. Within the family, PLA comprises the highest proportion in terms of annual production rate. However, there are several issues associated with the physicochemical properties of PLA-based implants [13]. The high hydrophobicity of PLA and local acidification caused by long-term material degradation decrease cellular attachment, cell infiltration of the scaffold, cell survival, and proliferation [14]. Blending PLA with some other polymers, including proteins, is a way to overcome these limitations.

Electrospinning enables the use of a diverse range of polymers as nonwoven mats. Electrospun nanofiber mats efficiently mimic the extracellular matrix, which makes them ideal tissue engineering scaffolds [15,16]. The aforementioned issues of PLA-based electrospun mats could be overcome by combining the polyester with biopolymers, for example, proteins (collagen, gelatin, elastin, and fibrinogen). Natural polymers promote cell adhesion by providing a hydrophilic environment and containing integrin-binding sites for cell adhesion molecules on the cell surface [17]. The simplest method to combine several polymers in fibrous materials is blend electrospinning. Nanofibrous mats comprising such polymer blends combine the biological properties of natural polymers with the mechanical strength and durability of synthetic polymers [18]. Usually, an increase in protein content increases the hydrophilicity [19,20] but negatively affects the mechanical strength of the electrospun material [21].

An increased biodegradation rate of electrospun mats along with maintaining their physicochemical, morphological, and mechanical properties can broaden their application as scaffold-like materials for tissue engineering. We demonstrated here that the addition of 10 wt% gelatin to the PLA electrospun mats insignificantly affects their physicochemical properties, such as hydrophilicity, mechanical characteristics, and in vitro degradation, while dramatically accelerates the in vivo biodegradation after subcutaneous implantation.

## 2. Results

### 2.1. Morphology

It was found using scanning electron microscopy (SEM) that both PLA and PLA–gelatin mats primarily comprise smooth cylindrical fibers (Figure 1A,C), although PLA–gelatin mats also contain rare flat ribbons (Figure 1C). The statistical processing of the SEM images has shown that the average diameters of PLA and PLA–gelatin fibers were 550 ± 300 nm (Figure 1B) and 380 ± 130 nm, respectively (Figure 1D).

### 2.2. FTIR Analysis

The FTIR spectra of the electrospun PLA, gelatin, and PLA–gelatin (9:1) mats are shown in Figure 2. The PLA mat spectrum contains the absorption band at 1753 cm^−1^ corresponding to the C=O stretching in carboxyl groups. The two bands at 1182 and 1086 cm^−1^ originate from symmetric and asymmetric valence vibrations of the C-O-C group. Symmetric and asymmetric deformation vibrations of C-H bonds in methyl groups produce two bands at 1454 and 1383 cm^−1^, respectively [22]. The gelatin characteristic bands were 1637 cm^−1^ (C-O stretching vibration), 1532 cm^−1^ (N-H bending vibration), 1451 cm^−1^ (C-H bending vibration), and 1333 cm^−1^ (C-N stretching vibration) [23]. To investigate the molecular interactions between PLA and gelatin, we examined the spectra of PLA–gelatin (9:1) and PLA–gelatin (1:1) blends. Both blends exhibit bands at approximately 1753 cm^−1^, which is a characteristic band of PLA (marked red in Figure 2). They also show bands approximately at 1637 and 1532 cm^−1^, which are attributed to gelatin (marked grey in Figure 2). Due to the small amount of gelatin, these bands in the PLA–gelatin (9:1) blend spectrum are weak but still distinct. Thus, the obtained data clearly show that PLA and gelatin do not undergo chemical modifications.

### 2.3. Wettability

Wettability is one of the important characteristics of biocompatible materials. Once hydrophobic surfaces are favorable for bacterial growth, hydrophilic surfaces are known to promote eukaryotic cell adhesion and proliferation [24]. Proteins are frequently added to the electrospinning solution to improve the wettability of the nonwoven material [19,20]. PLA is a hydrophobic polymer, and its blending with gelatin is expected to enhance hydrophilicity. The addition of gelatin to the electrospinning solution, however, slightly increased the contact angle from 129 ± 7° for the PLA to 135 ± 4° for PLA–gelatin mats and has not changed further during 40 s (Figure 3).

### 2.4. In Vitro Degradation Analysis

To evaluate in vitro degradation behavior of the obtained mats, we incubated the samples in PBS and Fenton’s reagent, which mimics oxidative stress caused by the inflammatory process after biomaterial implantation. The fragments of PLA and PLA–gelatin mats were incubated at 37 °C, which also reproduces physiological conditions.

The degradation rate was determined as a ratio of the weight loss to the initial sample weight. Upon 8 weeks of incubation, we observed a 10% weight loss (Figure 4). At the last timepoint, we observed a statistically significant difference in the weight loss between PLA and PLA–gelatin incubated in either PBS or in Fenton’s reagent.

The degradation process was accompanied by changes in microstructure. The SEM imaging showed that the fibers of both compositions acquired numerous constrictions (Figure 5 and Figure 6). Qualitatively, the degraded mats looked similar regardless of the chemical composition and degradation medium used.

### 2.5. Mechanical Properties

Table 1 displays the mechanical testing results. We investigated the mechanical properties of PLA and PLA–gelatin mats in dry and wet states. Dried PLA–gelatin mats exhibited a considerable drop in extensibility compared to PLA mats. Upon wetting, the extensibility of PLA mats reduced, while PLA–gelatin mats exhibited increased extensibility (elongation at break increased two times). In terms of mechanical strength, both types of mats have shown a similar decrease in upon contact with water.

### 2.6. Viability of Mammalian Cells Cultured on Electrospun Mats

We examined the influence of gelatin addition to the composition of PLA electrospun mats on the safety of this biomaterial for mammalian cells. First, we compared the 3T3 fibroblast proliferation rate on the surfaces of PLA, PLA–gelatin electrospun mats, and culture plastic. The obtained DNA profiles of 3T3 cells (Figure 7A) have shown that fibroblast cultivation on the electrospun mats does not affect the cell cycle or proliferation rate. We also did not reveal any differences between PLA and PLA–gelatin mats (Figure 7B). Then, we evaluated the cytotoxicity of these biomaterials for fibroblasts by simultaneous staining with propidium iodide (PI) and Hoechst dyes, followed by fluorescent imaging (Figure 7C). The image analysis indicated an insignificant number of dead cells, which were found in all samples. Overall, fibroblast cultivation on PLA and PLA–gelatin electrospun mats does not significantly affect cell viability in comparison with culture plastic (Figure 7D).

### 2.7. In Vivo Biodegradation

The biodegradation behavior of PLA and PLA–gelatin electrospun mats was evaluated in a mouse model after subcutaneous implantation. The degradation rate of PLA and PLA–gelatin electrospun discs was determined by measurement of their thickness before implantation and after engraftment on days 14 and 30 using histological images (Figure 8A).

The thinning of the implants occurred in both types of mats and was accompanied by the gradual replacement of the mat volume with connective tissue. The implanted mats did not induce a significant inflammatory response. At the same time, PLA–gelatin electrospun discs exhibited a greater infiltration with the host cells than PLA mats (Appendix A), which might be a result of their improved cell adhesion properties or higher immunogenicity due to the inclusion of bovine gelatin. Regardless, we observed a faster degree of biodegradation in the case of PLA–gelatin mats, resulting in a statistically significant 30% thinning after 1 month, whereas the thickness of PLA mats decreased by ~10% (Figure 8B).

## 3. Discussion

PLA is a biodegradable and biocompatible polymer that is widely used for different biomedical applications, including the production of suture threads and bone fixation screws [25]. It possesses high durability and low toxicity, although PLA is a hydrophobic and relatively long-term biodegradable material [26]. In this study, we analyzed the influence of a small addition of gelatin to electrospun PLA mats on their physicochemical and biological properties.

Although both the PLA and PLA–gelatin fibers have the same morphology (smooth surface and cylindrical shape), the mean fiber diameter decreases by ~30% with the addition of gelatin to the mats’ composition (Figure 1). Similar results were obtained for PCL and PCL–gelatin fibers, and the diameter of the fibers decreased with an increase in the proportion of gelatin in the composition [27]. Since gelatin is a polyelectrolyte, its addition can lead to an increased charge density of the blend as compared to the pure PLA solution. Thus, a decrease in the mean fiber diameter in blended mats may be caused by more efficient stretching of the polymer jet or (and) by the formation of branched fibers due to the increased charge density [28].

We observed that an addition of 10 wt% gelatin to the PLA mat slightly increased the contact angle. This result seems contradictory because the addition of a hydrophilic (and even water-soluble) component should decrease the contact angle. The observed effect can be explained qualitatively if we consider wetting hierarchy at multiple scales [29]. First, we can consider the interaction between the liquid and the material of the fibers in the case of an infinite substrate. Second, we can consider the interaction between the liquid and a nanofiber with a certain diameter. Third, we can consider the interaction between the liquid and the network of fibers, which includes the air-filled gaps. Obviously, the wetting contact angle of an electrospun mat strongly depends on the fiber diameter and porosity [30,31,32].

Gelatin should be evenly distributed throughout the fibers [33,34]. We expect that an individual PLA–gelatin fiber is more hydrophilic than the one made of pure PLA. This corresponds to the “lower” level of hierarchy described above. However, the addition of gelatin changes not only the interaction between water and the fiber material but also mean fiber diameter (380 ± 130 nm for PLA–gelatin versus 550 ± 300 nm for PLA) and, presumably, causes higher sample porosity. From the point of macro-scale wetting, these effects seem to be more important than the difference in the wetting of individual fibers.

We conducted mechanical testing of both dry and wet mats. Since the scaffolds are designed to function in contact with the physiological fluids, their properties in the wet state seem to be more important than in the dry state. There are some studies showing the decrease in mechanical properties of polyester materials upon the addition of gelatin. For example, the addition of gelatin into polycaprolactone (PCL) scaffolds drastically decreased their mechanical characteristics [1,35]. For this reason, scaffolds with high gelatin content usually require crosslinking [1,36]. The mechanical characteristics of both dried and wet PCL–gelatin mats were examined earlier [37]. Dried PCL–gelatin mats seem to be more extensible than PLA–gelatin mats (66.3 ± 5.3 vs. 22 ± 10), but this may be explained by the higher extensibility of electrospun PCL compared to electrospun PLA [38]. Nevertheless, PCL–gelatin mats also increased their extensibility after wetting. Dried mats demonstrated the same level of strength as the PLA–gelatin mats observed in the current work, although they became more fragile after wetting. However, in most of the experiments described here, PLA–gelatin mats exhibited a higher mechanical performance than PLA mats (Table 1). The differences were not statistically significant, but they indicate a lack of mechanical property deterioration after a small addition of gelatin. A single exclusion was found for dried PLA mats, which exhibited a higher elongation at break than PLA–gelatin counterparts.

We found that PLA and PLA–gelatin mats do not affect the proliferation and viability of the fibroblasts cultivated on their surface during a 72-h incubation (Figure 7). Long-lasting contact with body fluids involves chemical, physical, mechanical, and biological interactions between the material and the surrounding environment and results in the degradation of the material [13,39]. Implanted PLA-containing materials can degrade via a few presumable mechanisms. First, they undergo hydrolysis in a water medium [40]. Polymer hydrolysis stands for the destruction of polymers into oligomers and monomers due to the interaction of water with water-labile bonds. Second, ROS generated in the tissue environment by pro-inflammatory immune cells also could contribute to the oxidative degradation of the mats. Finally, PLA-containing mats can be degraded enzymatically in the presence of serine proteases [40], lipase [41], or alcalase [42]. We modeled hydrolytic and oxidative degradation by incubating the mats in PBS and Fenton’s reagent. The weight reduction in both PLA and PLA–gelatin mats was ~10% after 8 weeks, regardless of the incubation medium. However, gelatin incorporation into the mat slightly increased the degradation rate. In contrast with in vitro experiments, the addition of gelatin to PLA mats significantly increased in vivo biodegradation (Figure 8). There are several possible explanations for the results obtained. First, gelatin is a target for metalloproteinases (MMPs) [43], which can be released into the tissue microenvironment by macrophages and fibroblasts. Once gelatin constitutes 10 % of the composite mat, its enzymatic hydrolysis can accelerate overall degradation of the composite material. Next, gelatin is a ligand for integrin receptors that facilitates contact formation between the cells and scaffold fibers and cell infiltration of the biomaterial. As a result, it makes the fibers more accessible to ROS and enzymes generated by these cells.

## 4. Materials and Methods

### 4.1. Materials

Polylactide (PLA, M_W_~260,000) and gelatin were obtained from Sigma-Aldrich (St. Louis, MO, USA), 1,1,1,3,3,3-hexafluoroisopropanol (HFIP) was purchased from P&M-Invest (Moscow, Russia).

### 4.2. Electrospinning

The electrospinning solution was prepared by dissolving the dried PLA or PLA–gelatin mixture (9:1) in HFIP at 150 mg mL^−1^. The solution was aspirated into a 5-mL syringe equipped with a 21G metal needle. Electrospinning was performed using the Nanofiber Electrospinning Unit (Tong Li Tech, Shenzhen, China). The mats were produced using an accelerating voltage of 30 kV. The feed rate of the electrospinning solution was set at 2 mL per hour and was maintained by the syringe pump (KD Scientific, Holliston, MA, USA). The mats were electrospun using a rectangular frame collector (75 × 45 mm) positioned between the electrodes at a distance of 20 cm. The residual solvent was removed by drying in a vacuum cabinet.

### 4.3. Fourier Transform Infrared (FT-IR) Spectroscopy

To perform the FTIR analysis, PLA (150 mg mL^−1^), gelatin (10 wt%), PLA–gelatin (9:1, 150 mg mL^−1^), and PLA–gelatin (1:1, 150 mg mL^−1^) solutions in HFIP were electrospun according to the procedure described in Section 4.2. The chemical structure of the electrospun mats was analyzed using the Spectrum Two FT-IR Spectrometer (Perkin-Elmer, Waltham, MA, USA) with the MIRacle ATR unit (PIKE Technologies, Fitchburg, WI, USA). All the samples were pressed against a ZnSe prism and scanned in the range from 2000 to 500 cm^−1^. All spectra were collected by cumulating 30 scans per spectrum; the background signal was pre-registered.

### 4.4. Imaging of Mats Using SEM

The morphology of the electrospun mats was investigated using a scanning electron microscope, the Zeiss Merlin (Zeiss, Oberkochen, Germany), equipped with Gemini II Electron Optics. The samples were preliminarily coated with a 10 nm gold–palladium alloy using the Sputter Coater Q150T (Quorum Technologies, Louis, UK). The images were acquired at 2–5 kV acceleration voltage and 100–200 pA current. The FIJI software (NIH, Bethesda, MD, USA) was used to process SEM images [44].

### 4.5. Contact Angle Measurements

The wettability of the mats was assessed by measuring the contact angle (CA) between a water drop and the sample surface at room temperature with a DSA 25E instrument (Kruss, Hamburg, Germany). A 2 μL drop of water was placed onto the surface of the mats. The FIJI software was used to analyze the shape of the drop and calculate the contact angle.

### 4.6. Hydrolytic and Oxidative Degradation

A hydrolytic and oxidative degradation study was carried out in accordance with the ISO 10993-13 international standard. Hydrolytic degradation was performed by the mat’s incubation in phosphate buffered saline. The incubation in Fenton’s reagent (100 μM FeSO_4_ and 1 mM H_2_O_2_) was used to simulate the mutual effects of hydrolytic and oxidative degradation. Samples were preliminarily weighted using a Discovery analytical balance (Ohaus, Parsippany–Troy Hills, NJ, USA). Then, the samples of known weight (~10 mg) were placed in 500 μL of one of the mentioned incubation solutions. A partial substitution of the incubation solution (300 μL) was carried out every 7 days with pre-centrifugation to avoid the loss of the sample fragments. The centrifugation was performed using a MiniSpin centrifuge (Eppendorf, Germany) at 12,000 rpm for at least 10 min. After certain time intervals, the samples underwent triple washing via centrifugation, with the maximum possible volume of supernatant removed and the supernatant replaced with distilled water. The washing was followed by the removal of the maximum possible volume of supernatant and drying in a vacuum cabinet (Binder, Germany). The dried sample and its fragments were weighted, and the degradation degree was determined as the ratio of weight loss to the initial weight:(1)$$r=\frac{m_{0}-m}{m_{0}}$$
where *r* is a degradation degree, $m_{0}$ is an initial weight of the sample, and m is weight of the dry sample after incubation.

### 4.7. Mechanical Testing

To perform mechanical testing, the mats were cut into 5 × 45 mm samples using a laser engraving machine, the LaserPro Spirit GLS (GSS, Taipei City, Taiwan), equipped with a 100 W CO_2_ laser. The thickness of each sample was measured immediately before the experiment using a contact measurement gauge (Logitech, Lausanne, Switzerland). Tensile testing was carried out using the TA.XTPlus Connect testing machine (Texture Technologies Corp. and Stable Micro Systems, Godalming, UK). The ultimate elongation at break and ultimate stress were determined for dry and wet (after 30-min incubation in PBS) samples by uniaxial stretching at a linear speed of 5 mm min^−1^.

### 4.8. Cell Culture

Embryonic mouse fibroblasts NIH3T3 (ATCC^®^ CRL-1658™) (hereafter, 3T3 fibroblasts) were cultured in DMEM growth medium supplemented with 10% fetal bovine serum, penicillin/streptomycin, and a solution of non-essential amino acids (all from Gibco). All cultured cells were grown at 37 °C in a humidified 5% CO_2_ atmosphere.

### 4.9. Analysis of DNA Profiles and Cell Viability

PLA and PLA–gelatin electrospun mats with a mean diameter of 35 mm were treated with UV-C light for 10 min on each side, then wetted with a 20% ethanol solution and washed in sterile mQ water. After washing, the wetted mats were placed on the bottom of 6-well culture plates. Cell suspensions of NIH3T3 fibroblasts were seeded on the mats (3 × 10^5^ cells per well) in a volume of 200 μL. After an hour of incubation at 37 °C in humidified 5% CO_2_ atmosphere, an additional 1.3 mL of fresh growth medium per well was gently added, followed by 72-h cultivation. Then, the mats were washed with Versene solution, and the cells were removed using a 0.25% trypsin solution. Fibroblasts were collected after trypsinization by centrifugation at 3500 rpm for 4 min, washed with Versene solution, and fixed in 66% ethanol. After fixation, the cells were incubated in PBS containing 1 μg mL^−1^ propidium iodide (PI) (Thermo Fisher Scientific, Waltham, MA, USA) and 50 U mL^−1^ RNAse A (Thermo Fisher) at room temperature for 30 min. DNA profiles were determined by collecting 10,000 events in the PI channel using the cytometer CytoFLEX (Beckman Coulter, Brea, CA, USA).

To determine the viability of the cells grown on the surface of electrospun mats, NIH3T3 fibroblasts were seeded on PLA and PLA–gelatin membranes with a mean diameter of 13 mm placed on the bottom of a 24-well plate at a density of 150,000 cells per well in a volume of 100 μL. After an hour of incubation at 37 °C in humidified 5% CO_2_ atmosphere, an additional 400 μL of fresh growth medium per well was gently added, followed by a 2-day cultivation. Then, the mats were washed with HBSS, followed by staining with propidium iodide (final concentration of 1 μg mL^−1^) and Hoechst 33342 (final concentration of 1 μg mL^−1^) for 10 min at room temperature. The samples were photographed using a fluorescent microscope (AxioVert. A1, Zeiss, Oberkochen, Germany) equipped with a ×20/0.6 objective lens. The cells stained with PI and Hoechst 33342 were defined as dead, whereas the cells stained with Hoechst 33342 only were defined as viable. Three random areas with at least 200 cells each were analyzed.

### 4.10. In Vivo Biodegradation of Electrospun Mats

To examine the biodegradation behavior of electrospun mats, PLA and PLA–gelatin discs with a mean area of 1.75 cm^2^ and a mean width of 134 ± 10 and 153 ± 11 µm, respectively, were implanted subcutaneously on the backs of 8-week-old female C57black/6 mice (Stolbovaya, Moscow region, Russia). The thickness of mats was measured immediately before the experiment using a contact measurement gauge (Logitech, Lausanne, Switzerland). After implantation, the incised skin was sutured, and the skin area over the implanted disc was outlined with a permanent marker. During the procedure, the animals were anesthetized by intraperitoneal injection of 320 mg kg^−1^ of avertin in Hanks’ balanced salt solution.

On days 14 and 30 after implantation of electrospun mats, 6 mice from each group were anesthetized to undergo transcardial perfusion with 50 mL of PBS and 50 mL of 10% buffered neutral formalin (Thermo Scientific). Tissues with implanted mats were excised and incubated for 5 days in 10% buffered neutral formalin, followed by embedding into paraffin wax. Then, the tissue blocks were cut into 5 μm thickness slices and stained with haematoxilyn-eosin. Tissues were photographed using an inverted Olympus CX41 (Olympus Co., Tokyo, Japan) equipped with a UPlanApo 20×/NA 0.70 objective lens.

## 5. Conclusions

We demonstrated that the small addition of gelatin (10%) to the electrospun PLA mats accelerates in vivo degradation without a significant decrease in the mechanical properties of wet mats. Even though we observed a slight decline in the hydrophilicity of the blended mats, it did not affect their biocompatibility. We believe that these findings can be used for the biocompatibility and biodegradation improvement of the polyester tissue engineering scaffolds.

## Figures and Tables

**Figure 1 ijms-24-03535-f001:**
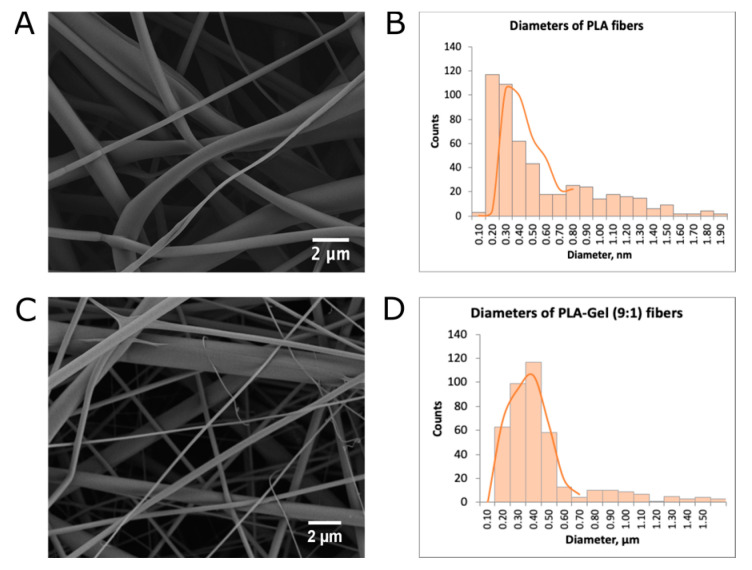
SEM analysis of electrospun mat morphology. (**A**) SEM image of an electrospun PLA mat. (**B**) Fiber diameter distribution obtained from the SEM image analysis of PLA mats. (**C**) SEM image of an electrospun PLA–gelatin (9:1) mat. (**D**) Fiber diameter distribution obtained from the SEM image analysis of PLA–gelatin (9:1) mats.

**Figure 2 ijms-24-03535-f002:**
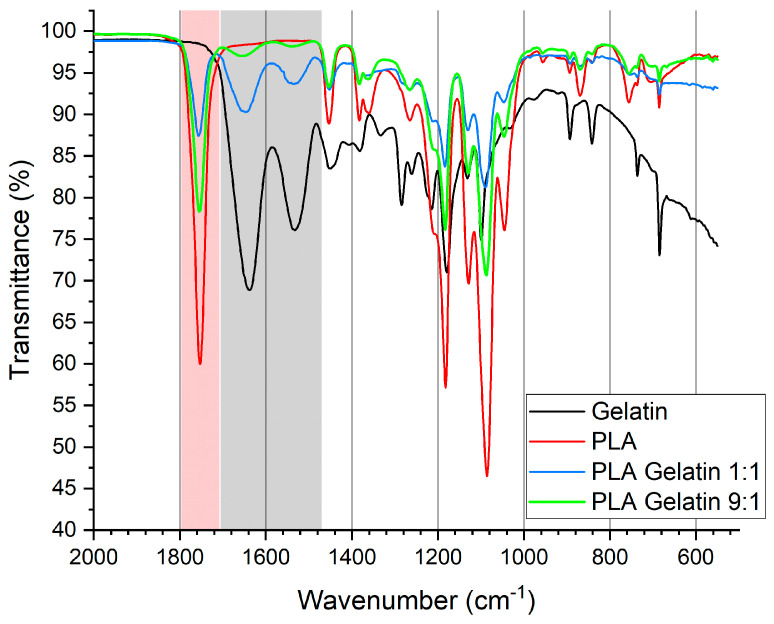
FTIR spectra of electrospun gelatin, PLA, PLA–gelatin (1:1) blend, and PLA–gelatin (9:1) blend.

**Figure 3 ijms-24-03535-f003:**
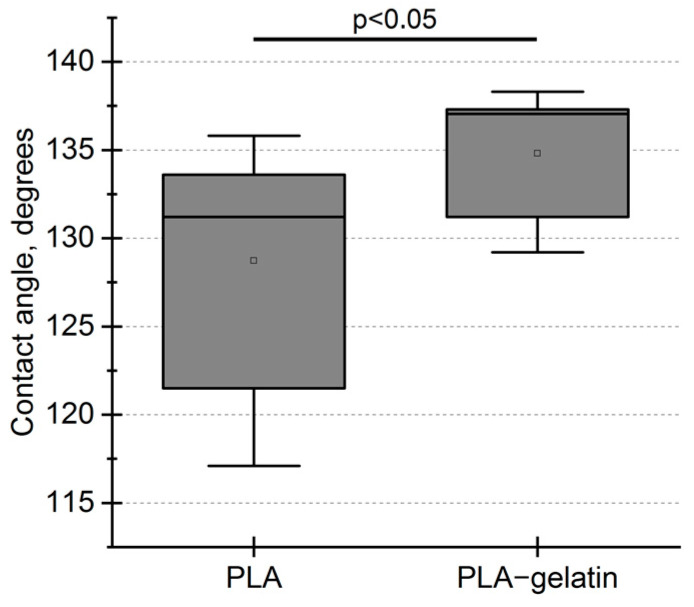
Initial contact angle of electrospun PLA and PLA–gelatin (9:1) mats. No water absorption was observed during the experiment (40 s). According to the Mann–Whitney U-test, the difference is statistically significant (*p* < 0.05).

**Figure 4 ijms-24-03535-f004:**
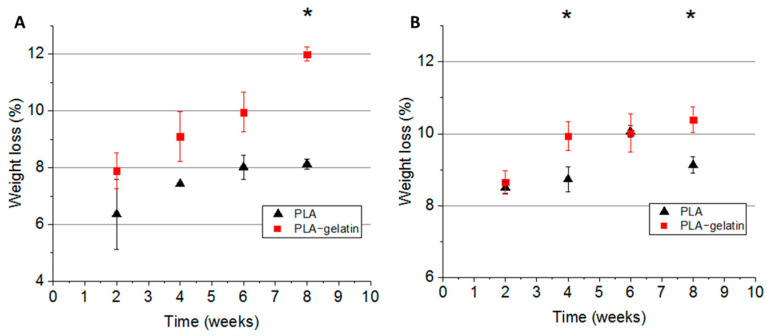
Degradation rate (weight loss over time) of electrospun PLA and PLA–gelatin (9:1) mats over 8 weeks of incubation in PBS (**A**) and in Fenton’s reagent (**B**) at 37 °C. The asterisk (*) marks statistically significant differences (*p* < 0.05) between PLA and PLA–gelatin samples according to Student’s *t*-test.

**Figure 5 ijms-24-03535-f005:**
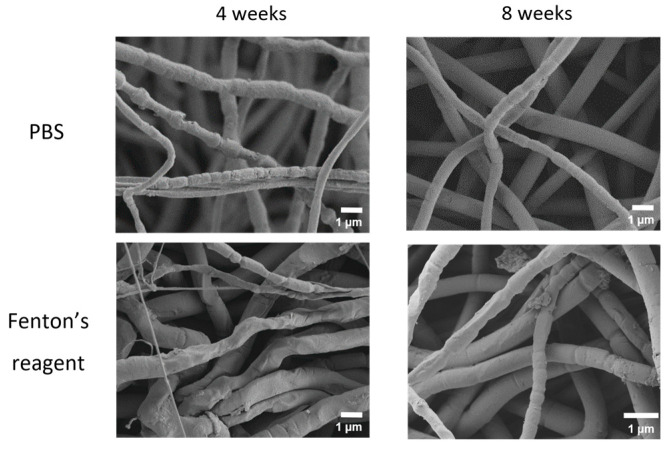
SEM images of electrospun PLA mats after 4 and 8 weeks of incubation in PBS and in Fenton’s reagent.

**Figure 6 ijms-24-03535-f006:**
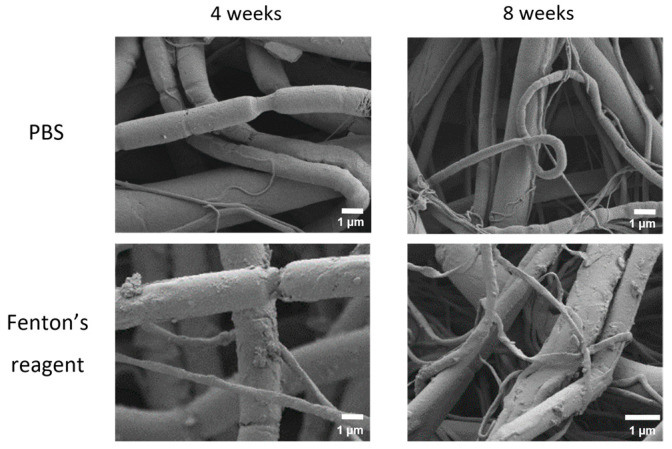
SEM images of electrospun PLA–gelatin (9:1) mats after 4 and 8 weeks of incubation in PBS and in Fenton’s reagent.

**Figure 7 ijms-24-03535-f007:**
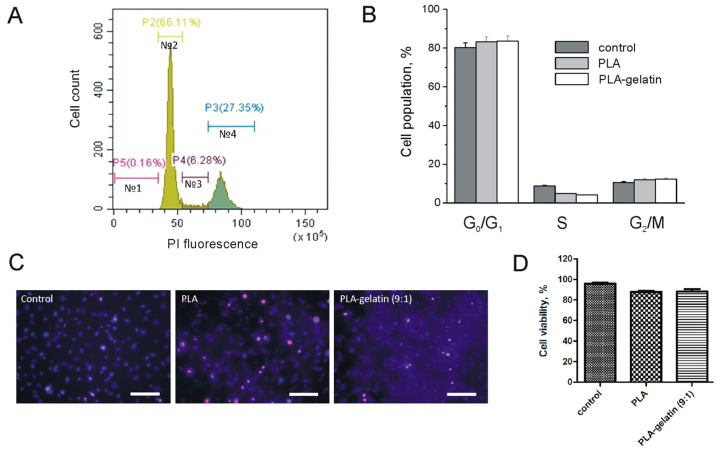
Analysis of proliferation rate and viability of 3T3 fibroblasts cultured on PLA and PLA–gelatin electrospun mats. (**A**) A typical view of the DNA profile of the 3T3 cell population, where zones 1, 2, 3, and 4 correspond to the apoptotic, G0/G1-phase, S-phase, and G2/M-phase cell fractions. (**B**) Fractions of 3T3 cells, cultured on PLA, PLA–gelatin electrospun mats, and on culture plastic (control). (**C**) Merged photographs of 3T3 fibroblasts, cultured on different materials, after staining with Hoechst 33342 (blue cell nuclei) and PI (red cell nuclei). The scale bar is 100 μm. (**D**) Viability of the fibroblasts, cultured on PLA, PLA–gelatin electrospun mats, and on culture plastic (control).

**Figure 8 ijms-24-03535-f008:**
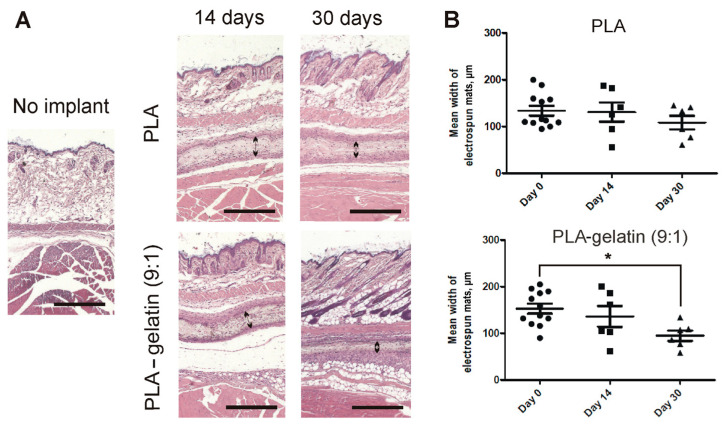
In vivo biodegradation analysis of PLA and PLA–gelatin electrospun mats. (**A**) Histological images of PLA and PLA–gelatin discs on days 14 and 30 after subcutaneous implantation in C57black/6 mice. A double-ended arrow indicates the thickness of the implanted biomaterial. The scale bar is 500 μm. (**B**) Change in electrospun mat thickness over time. The data are presented as mean±SD. * *p* < 0.05 one-way ANOVA followed by a post hoc Dunnett’s *t*-test.

**Table 1 ijms-24-03535-t001:** The mechanical testing parameters of electrospun PLA and PLA–gelatin (9:1) mats.

Parameter	State	PLA	PLA–Gelatin (9:1)
Elongation at break, %	Dry	56 ± 16	22 ± 10
	Wet	44 ± 12	45 ± 7
Maximum stress, MPa	Dry	3.5 ± 0.4	4.4 ± 0.4
	Wet	3.4 ± 0.3	3.9 ± 0.4

## Data Availability

The data presented in this study are available on request from the corresponding author.

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
