# Peer review of "Acceleration of Electrospun PLA Degradation by Addition of Gelatin"

_ijms, 2023, doi:10.3390/ijms24043535_

Round 1
Reviewer 1 Report
This work is about the addition of a small amount of gelatin to cause degradation in the electrospun PLA. There are studies on this subject in the literature, the manuscript is well structured and the concepts are clear. However there are some analyzes to improve:
- PLA-gelatin samples must be characterized (for example by FTIR)
- The authors present 10 wt% of content in gelatin, how did the authors come up with this value? Have other values been used? Did they do an optimization?
- To complete the study the authors should address the antibacterial activity
Author Response
- PLA-gelatin samples must be characterized (for example by FTIR)
We have conducted an experiment using FTIR spectroscopy. The obtained results have been added to the Results section.
- The authors present 10 wt% of content in gelatin, how did the authors come up with this value? Have other values been used? Did they do an optimization?
The aim of the current work was to characterize this specific composition. We have tried different compositions of PLA and gelatin, but the one described in the manuscript seems the most promising. The other PLA-gelatin compositions were used in some of our previous works, for example PLA:gelatin (80:20) was used in dye adsorption studies (10.1080/15421406.2018.1563945) and PLA:gelatin (50:50) was used in the studies of membrane homogeneity (10.1063/1.5087680). During the previous research, we faced several issues associated with the PLA-gelatin blends, for example, the weak mechanical properties caused by the high gelatin contents.
- To complete the study the authors should address the antibacterial activity
The investigated biomaterials did not have antibacterial activity. Prior to the in vitro and in vivo biocompatibility studies, the electrospun mats were sterilized by UV irradiation as described in the updated version of the manuscript (Materials and methods, page 10, lines 341-343).
Reviewer 2 Report
Alexandra Bogdanova et al. blended small amount of gelatin to PLA during electrospinning, and further explored the addition of gelatin on both the in vitro and in vivo degradation of as-prepared electrospun nanofibres. Some major concerns and issues should be addressed before publication.
1. The Abstract section should be rewritten in a better and clear way. Some important result data should be presented in this section.
2. Please state the merits and demerits of PLA, compared with some other biopolymers like PCL, PGA, PLGA, and PPDO, etc, in the introduction section. Moreover, some recent review work about electrospinning like 10.3390/ijms19030745, and 10.1016/j.mtchem.2022.100944 are recommended to be added.
3. Please state the reasons why 10 percent gelatin was chosen, not 1%, 5%, or 15%, etc. Are there any preliminary experiment conducted?
4. Please explain why the addition of gelatin could dramatically decrease the fiber diameter of as-prepared nanofibers.
5. The actual photographs of water contact angle of different samples are suggested to be added in Figure 2, in order to strengthen the readability. Moreover, please justify the reasons why the addition of gelatin led to the increase of hydrophobic performance. It seems nonsense.
6. For mechanical test, why was the time point of 7 days selected, not 4 weeks or 8 weeks? Moreover, the Young’s modulus of each sample are suggested to be added in Table 1.
7. Does it have any significant different in Figure 5B and D?
8. The authors claimed that PLA-gelatin electrospun discs exhibited a greater infiltration with the host cells than PLA mats during the in vivo study, but it’s really difficult to figure out this conclusion for the reviewer. Please add some more detailed data and explanation.
9. Blending gelatin and PLA (10.3390/nano12010006; 10.3389/fbioe.2021.684105; 10.1016/j.apmt.2022.101542) have been widely investigated to fabricate electrospun nanofibers, so some more descriptions should be added to compare and discuss the previous works in the discussion section.
10. In the Materials and Methods section, some important information were missing. For instant, how about the molecular weight of employed PLA and electrospinning parameters? The testing parameters for mechanical characterization should be given.
11. The grammar and writing should be improved in the whole manuscript.
Author Response
- The Abstract section should be rewritten in a better and clear way. Some important result data should be presented in this section.
As you suggested, we have rewritten the Abstract section.
- Please state the merits and demerits of PLA, compared with some other biopolymers like PCL, PGA, PLGA, and PPDO, etc, in the introduction section. Moreover, some recent review work about electrospinning like 10.3390/ijms19030745, and 10.1016/j.mtchem.2022.100944 are recommended to be added.
We selected PLA for this study for two reasons. First, it is one of the most slow-degrading polyesters and the acceleration of its degradation is a relevant task. Second, from the standpoint of chemistry, it is the simplest polyester, and so it is a convenient model for the whole family of biomedical polyesters. We highlighted these two points in the updated version of the manuscript.
We have cited the two articles mentioned by the reviewer in the Introduction.
- Please state the reasons why 10 percent gelatin was chosen, not 1%, 5%, or 15%, etc. Are there any preliminary experiment conducted?
The aim of the current work was to characterize this specific composition. We have tried different compositions of PLA and gelatin, but the one described in the manuscript seems the most promising. The other PLA-gelatin compositions were used in some of our previous works, for example PLA:gelatin (80:20) was used in dye adsorption studies (10.1080/15421406.2018.1563945) and PLA:gelatin (50:50) was used in the studies of membrane homogeneity (10.1063/1.5087680). During the previous research, we faced several issues associated with the PLA-gelatin blends, for example, the weak mechanical properties caused by the high gelatin contents.
- Please explain why the addition of gelatin could dramatically decrease the fiber diameter of as-prepared nanofibers.
Unfortunately, the detailed molecular explanation of this phenomenon is not known. We hypothesize that addition of gelatin increases the charge density of the surface of the polymer jet. The increase in charge density occurs because of the intense ionization of gelatin molecules; it intensifies the stretching of the jet and can lead to branching - both these processes manifest as a decrease in fiber diameter. A similar explanation is discussed in the article (10.1016/j.msec.2012.12.015), which regards a PCL/gelatin blend.
- The actual photographs of water contact angle of different samples are suggested to be added in Figure 2, in order to strengthen the readability. Moreover, please justify the reasons why the addition of gelatin led to the increase of hydrophobic performance. It seems nonsense.
The unexpected counter-intuitive influence of gelatin on the observed contact angle might originate from the lower diameter of the fibers in the PLA-gelatin membrane and (presumably) higher porosity. We updated the corresponding paragraph in the Discussion section to emphasize this point.
- For mechanical test, why was the time point of 7 days selected, not 4 weeks or 8 weeks? Moreover, the Young’s modulus of each sample are suggested to be added in Table 1.
When we carried out the experiments, we used the 7-day point to estimate the trend of mechanical changes over time. This point did not match the other timepoints used in the study (e.g., 4 weeks or 8 weeks). Thus, we decided to remove the results of mechanical testing after a 7-day incubation to improve the overall readability.
Unfortunately, our mechanical testing device does not output stress/strain relation curves, so we can’t calculate Young's moduli.
- Does it have any significant different in Figure 5B and D?
There were no significant differences in cell population and cell viability between control, PLA and PLA-gelatin mats. It means that fibroblast cultivation on PLA and PLA-gelatin electrospun mats does not significantly affect cell viability in comparison with culture plastic (page 6, lines 161-163).
- The authors claimed that PLA-gelatin electrospun discs exhibited a greater infiltration with the host cells than PLA mats during the in vivo study, but it’s really difficult to figure out this conclusion for the reviewer. Please add some more detailed data and explanation.
We added a reference to Supplementary Figure S1, which represents the increased infiltration of PLA-gelatin mats with the host cells as compared to PLA counterparts.
- Blending gelatin and PLA (10.3390/nano12010006; 10.3389/fbioe.2021.684105; 10.1016/j.apmt.2022.101542) have been widely investigated to fabricate electrospun nanofibers, so some more descriptions should be added to compare and discuss the previous works in the discussion section.
We have revised and rewritten the Discussion section in a clearer manner.
- In the Materials and Methods section, some important information were missing. For instant, how about the molecular weight of employed PLA and electrospinning parameters? The testing parameters for mechanical characterization should be given.
We have updated the Materials and methods section with the details.
- The grammar and writing should be improved in the whole manuscript.
Our manuscript has been thoughtfully revised a native speaker to improve readability.
Reviewer 3 Report
The manuscript is dedicated to the synthesis and study of PLA-gelatin fibers with the emphasis on their biodegradation ability in relation with mechanical properties. The manuscript contains enough novelty and scientifically relevant results and thus, in my opinion can be published.
One comment about viability of cells. My exerience tells me, that very likely there should be difference between fiber mats and flat plastic just due to the surface roughness. Therefore, when I see experiments in which either (almost) everything is alive or (almost) everything is dead in all samples, it raises question - how well were established conditions of experiments? Durations? Concentrations? Couldn't it happen that in better chosen conditions the samples would have differentiated behaviour? A comment on how conditions of this experiment were established would improve quality of the manuscript.
Author Response
One comment about viability of cells. My exerience tells me, that very likely there should be difference between fiber mats and flat plastic just due to the surface roughness. Therefore, when I see experiments in which either (almost) everything is alive or (almost) everything is dead in all samples, it raises question - how well were established conditions of experiments? Durations? Concentrations? Couldn't it happen that in better chosen conditions the samples would have differentiated behaviour? A comment on how conditions of this experiment were established would improve quality of the manuscript.
We observed a slight difference in proliferation rate and viability between fiber mats and culture plastic, but it was not statistically significant as shown in Figure 7. We provided a more detailed description of these experiments in an updated version of the manuscript (Materials and methods, page 10, lines 340-366).
Round 2
Reviewer 1 Report
The revised manuscript is ok for me
Reviewer 2 Report
The reviewer's comments have been addressed.